# Kuro Siwo: 33 billion $m^2$ under the water
# A global multi-temporal satellite dataset for rapid flood mapping

**Nikolaos Ioannis Bountos**[1,2*]
bountos@noa.gr

**Maria Sdraka**[1,2*]
masdra@noa.gr

**Angelos Zavras**[1,2]
azabras@noa.gr

**Ilektra Karasante**[1]
ile.karasante@noa.gr

**Andreas Karavias**[1]
karavias@hua.gr

**Themistocles Herekakis**[1]
therekak@noa.gr

**Angeliki Thanasou**[1]
thanasoua@gmail.com

**Dimitrios Michail**[2]
michail@hua.gr

**Ioannis Papoutsis**[1]
ipapoutsis@noa.gr

[1] **Orion Lab**
National Observatory of Athens & National Technical University of Athens

[2] **Harokopio University of Athens**

## Abstract

Global flash floods, exacerbated by climate change, pose severe threats to human life, infrastructure, and the environment. Recent catastrophic events in Pakistan and New Zealand underscore the urgent need for precise flood mapping to guide restoration efforts, understand vulnerabilities, and prepare for future occurrences. While Synthetic Aperture Radar (SAR) remote sensing offers day-and-night, all-weather imaging capabilities, its application in deep learning for flood segmentation is limited by the lack of large annotated datasets. To address this, we introduce Kuro Siwo, a manually annotated multi-temporal dataset, spanning 43 flood events globally. Our dataset maps more than 338 billion $m^2$ of land, with 33 billion designated as either flooded areas or permanent water bodies. Kuro Siwo includes a highly processed product optimized for flash flood mapping based on SAR Ground Range Detected, and a primal SAR Single Look Complex product with minimal preprocessing, designed to promote research on the exploitation of both the phase and amplitude information and to offer maximum flexibility for downstream task preprocessing. To leverage advances in large scale self-supervised pretraining methods for remote sensing data, we augment Kuro Siwo with a large unlabeled set of SAR samples. Finally, we provide an extensive benchmark, namely BlackBench, offering strong baselines for a diverse set of flood events globally. All data and code are published in our Github repository: https://github.com/Orion-AI-Lab/KuroSiwo.

## 1 Introduction

There is compelling evidence of compound effects that link extreme natural disasters worldwide [66]. The Intergovernmental Panel on Climate Change (IPCC) [72] underscores that floods represent

---

*Equal contribution

38th Conference on Neural Information Processing Systems (NeurIPS 2024) Track on Datasets and Benchmarks.

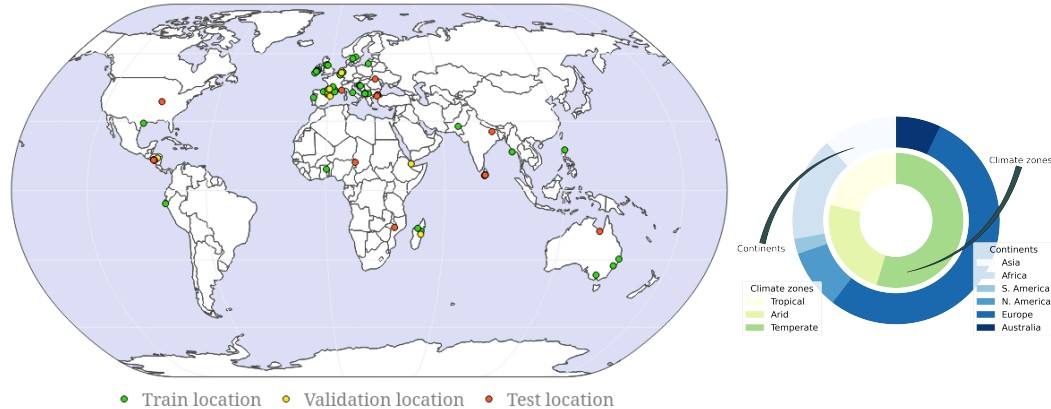

Figure 1: Spatial distribution of Kuro Siwo events.The test dataset encompasses flood events from entirely unseen locations on Earth.

the most frequent natural hazard globally and are projected to increase in frequency and intensity due to global warming [50, 6, 68]. The causes of flash floods, and more significantly, their expected impacts, exhibit significant variations contingent upon the exposure potential and vulnerabilities inherent in the affected population [77, 37] and socio-environmental assets [54]. In fact, floods tend to disproportionately affect the most impoverished and vulnerable segments of our society [67, 70, 11]. Research has revealed that approximately 170 million individuals exposed to substantial flood hazards are struggling with extreme poverty [67].

Recent tragic incidents in Pakistan [81, 82] and New Zealand [28] exemplify the gravity of flash floods, resulting in significant human casualties and substantial economic losses. According to the World Bank [83], the latest flood in Pakistan affected a staggering 33 million people, resulting in over $1,739$ fatalities. The mismanagement of disasters triggered by floods in itself carries substantial risks. For instance, stagnant floodwaters, which present a significant hazard for the transmission of water-borne and vector-borne diseases, have left more than 8 million people in a state of health crisis [83] with over 4 million children still living near contaminated floodwaters [81].

Remote sensing (RS) provides an opportunity for systematic, rapid and accurate flood mapping [14], crucial for effective flood impact management. Accurate flood mapping streamlines disaster response and relief efforts, allowing emergency responders, humanitarian organizations, and government agencies to efficiently allocate resources, deliver aid, and support affected populations [36]. It plays a crucial role in safeguarding critical infrastructure by identifying vulnerable assets [53] and assists in assessing flood risks, guiding urban planning [23], land-use zoning [61, 80], and formulating flood mitigation strategies [42]. Accurate flood maps are indispensable for insurance companies and financial institutions in evaluating flood-related risks, and managing financial exposure [76, 52, 78]. In the face of intensifying climate change impacts, flood mapping is crucial for adapting to evolving environmental conditions [55]. Lastly, it contributes to scientific research, enhancing our understanding of flood dynamics [49] and facilitating more informed decision-making in the long run [5, 19].

The exploitation of satellite data for flood mapping has seen extensive use [10], particularly with the emergence of Sentinel satellite platforms [79]. Specifically, the Sentinel-2 multispectral and Sentinel-1 Synthetic Aperture Radar (SAR) missions offer global, detailed and frequent imaging of the Earth's surface. However, high precipitation and extensive cloud cover during flood events impact multispectral sensors which effectively become blind and unreliable for operational scenarios. In contrast, SAR operates seamlessly day and night, unaffected by weather conditions, making it better suited for operational flood mapping. Thus, our research focuses on developing methods that can rapidly map flooded areas using SAR data.

Modern computer vision methods have already shown great promise in multiple RS applications [74, 75, 9, 62]. However, to the best of our knowledge, no machine learning method has notably excelled in both robust performance and generalization across diverse global flood events. We identify two primary reasons for this. Firstly, the task is inherently complex due to the presence of speckle noise in SAR [43], manifesting as random brightness variations in backscatter imagery. Second, deep

Table 1: Public datasets for flood mapping. S1 stands for Sentinel-1, S2 for Sentinel-2, GF2 for GaoFen-2, GF3 for GaoFen-3, UAV for Unmanned Aerial Vehicle, DEM for Digital Elevation Model and OSMh for Openstreet Map hydrography. Annotation methods are abbreviated as follows: A - Automatic, SA - Semi-automatic, M - Manual, CEMS - Copernicus Emergency Management Service. We assume the 7 continent scheme.

| Dataset | Sources | Imagery included? | SLC | GSD | # samples | sample size | # events | # events in test set | Timestamps | # continents | # classes | Annotation method |
|---|---|---|---|---|---|---|---|---|---|---|---|---|
| UNOSAT [60] | S1 | ✓ | ✗ | 10m | 58,128 | 256x256 | 15 | all + 1 new | Post | 2 | 2 | SA |
| SEN12-FLOOD [65] | S1, S2 | ✓ | ✗ | 10m | 336 | 512x512 | - | - | Time series | 3 | 2 | CEMS |
| Sen1Floods11 [8] | S1, S2 | ✓ | ✗ | 10m | 4,831 | 512x512 | 11 | 1 | Post | 5 | 3 | M, A |
| Ombria [22] | S1, S2 | ✓ | ✗ | 10m | 1,688 | 256x256 | 23 | all | Pre, Post | 5 | 2 | CEMS |
| WorldFloods [51] | S2 | ✓ | ✗ | 10m | 185,574 | 256x256 | 119 | 6 | Post | 6 | 3 | SA |
| GF-FloodNet [91] | GF2, GF3 | ✓ | ✗ | 1.5 - 5m | 13,388 | 256x256 | 8 | all + 3 new | Post | 4 | 2 | SA |
| Global Flood Database [77] | MODIS | ✗ | ✗ | 250m | 12,719 | - | 913 | - | Post | 6 | 2 | A |
| S1GFloods [71] | S1 | ✓ | ✗ | - | 5,360 | 256x256 | 46 | all + 2 new | Pre, Post | 6 | 2 | SA |
| MMFlood [56] | S1, DEM, OSMh | ✓ | - | 20m | 1,748 | 2000x2000 | 95 | 34 | Post | 6 | 2 | CEMS |
| RAPID-NRT [89] | S1 | ✓ | ✗ | 10m | 559 | - | 4 | - | - | 1 | 2 | A |
| CAU-Flood [31] | S1, S2 | ✓ | ✗ | 10m | 18,302 | 256x256 | 18 | 2 | Pre, Post | 4 | 2 | M |
| FloodNet [64] | UAV | ✓ | - | 1.5cm | 2,343 | 4000x3000 | 1 | 1 | Post | 1 | 9 | SA |
| ETCI2021 [59] | S1 | ✓ | ✗ | 20m | 66,810 | 256x256 | 5 | 2 | Post | 3 | 2 | - |
| **Kuro Siwo (labelled)** | **S1, DEM** | **✓** | **✓** | **10m** | **67,490** | **224x224** | **43** | **10** | **2 Pre, Post** | **6** | **3** | **M** |
| **Kuro Siwo (unlabelled)** | **S1, DEM** | **✓** | **✓** | **10m** | **466,357** | **224x224** | **43** | **10** | **2 Pre, Post** | **6** | **3** | **M** |

learning methods require a substantial amount of well-curated training data to fulfill their potential. While the Sentinel-1 mission offers abundant data for leveraging these methods in flood mapping, a comprehensive and well-annotated dataset is currently absent. This phenomenon is magnified when considering both the phase and amplitude information of SAR data.

To overcome this barrier, we curate time series data from Sentinel-1 SAR imagery linked to flood events worldwide, and manually annotate them with the expertise of SAR specialists. We provide two SAR products paired with the resulting reference annotation maps: Ground Range Detected (GRD) SAR with preprocessing tailored for flood mapping and Single Look Complex (SLC) SAR data with minimal preprocessing, containing both phase and amplitude signal. Additionally, we incorporate unlabeled SAR samples, to facilitate the exploration of semi-supervised and large-scale self-supervised learning (SSL). We name the resulting dataset "Kuro Siwo"[2] and release it publicly with the aim to propel research in this field towards developing operational tools that support civil protection authorities for the greater good.

Our main contributions can be summarized as follows:

- We publish Kuro Siwo, a global, manually annotated multi-temporal SAR dataset providing labels for 43 distinct flood events. Kuro Siwo consists of time series SAR imagery with dual polarization paired with elevation information. Its sheer size, meticulous annotation process and diversity make it a unique and highly valuable dataset for rapid flood mapping.

- Kuro Siwo is the first flood mapping dataset to provide both and coregistered GRD and SLC SAR data, offering the opportunity to leverage both amplitude and phase information.

- We enhance Kuro Siwo with a large unlabeled SAR dataset, facilitating global representation learning alongside a well-defined downstream task for evaluation. The final dataset contains more than 1.6 million SAR images grouped into more than 533, 000 time series.

- We publish the first benchmark on Kuro Siwo, namely BlackBench, offering diverse and strong baselines. BlackBench demonstrates the quality of Kuro Siwo, enabling the training of models that achieve performance higher than 80%, 78% and 83% F1-Score for flood, permanent and general water detection respectively on a challenging and diverse test set.

## 2 Related Work

### 2.1 Flood mapping datasets

In this section we summarize previous work on the creation of analysis-ready satellite based datasets for flood mapping. Our investigation is focused on three key aspects. First, the provided satellite sensors, second the ground truth generation process and finally the evaluation scheme. Please refer to Tab. 1 for an overview of our analysis.

---

[2]Named after a poem by sailor-poet Nikos Kavvadias (1910-1975).

Several datasets, such as SEN12-FLOOD, Sen1Floods11, Ombria, GF-FloodNet and CAU-Flood provide aligned multisource satellite imagery, facilitating the utilization of multiple modalities. A great number of datasets like UNOSAT, Sen1Floods11, WorldFloods, GF-FloodNet, MMFlood, ETCI2021 and FloodNet provide solely the post-flood acquisition omitting pre-event information which could greatly assist the mapping process and alleviate false positive predictions on permanent water bodies. In addition, databases like the Global Flood Database and RAPID-NRT comprise only flood mappings with no corresponding input imagery.

Given the difficulty and cost of satellite imagery photointerpretation, the creation of ground truth masks for large-scale datasets is not trivial, with many works offering automatic or semi-automatic methods to alleviate the need for a laborious labelling process (Tab. 1). Some datasets, like the Global Flood Database and RAPID-NRT, employ automatic annotation methods, introducing noise and inaccuracies in the labels due to their reliance on input data quality. Others, such as UNOSAT, WorldFloods, GF-FloodNet, S1GFloods, and FloodNet, utilize a semi-automatic pipeline where automatically generated labels undergo refinement by human annotators. It's worth noting that in Sen1Floods11, only $446$ samples from a single event were manually annotated, while the rest used a thresholding method without human intervention. Manual labeling through meticulous photointerpretation produces higher-quality ground truth mappings. Nevertheless, this approach is expensive, requiring expert input and potentially introducing annotator bias. Such labels have been produced for the CAU-Flood dataset to generate binary flood/no flood masks.

The proposed evaluation scheme is crucial for assessing the generalization capabilities of any flood mapping method. However, in Sen1Floods11, WorldFloods, CAU-Flood, ETCI2021 and FloodNet, a limited number of events are reserved for testing, potentially providing an insufficient indicator of a model's performance on unseen conditions. Additionally, in UNOSAT, Ombria, GF-FloodNet, and S1GFloods, samples from all flood events are shuffled and randomly split into training, validation, and test sets, posing a risk of data leakage where samples from the same area and satellite capture may appear in both training and test sets. To address this concern, the authors of UNOSAT, GF-FloodNet, and S1GFloods have also performed evaluation on imagery from a small number of new, unseen flood events, though these events are not included in the published datasets.

To our knowledge Kuro Siwo is the first flood mapping dataset offering ground truth labels based on expert photointerpretation at such an unprecedented scale ($43$ events), with wide spatial coverage spanning $6$ out of $7$ continents and $3$ out of $4$ major climate zones. Kuro Siwo can be the first step towards unlocking the true potential of deep learning methods for rapid flood mapping.

It is worth noting that phase information of SAR imagery has been consistently overlooked by the community regardless of the application. However, the phase signal can be used to generate useful byproducts, such as interferometric coherence that is suited for change detection applications [57] and especially for flood mapping [63]. In fact, S1SLC_CVDL [2] and OpenSarShip2.0 [44] are, to our knowledge, the only existing substantial datasets offering SLC data, both addressing tasks unrelated to floods. Kuro Siwo stands out as the only large-scale dataset that not only offers time series of SLC data paired with high-quality annotations but also provides wide spatiotemporal coverage with $43$ flood events from 2015 till 2022.

## 2.2 Deep learning for rapid flood mapping from SAR imagery

Several studies have explored the synergistic use of SAR and multispectral imagery, aiming to capitalize on the strengths of both modalities. In works such as [65] and [22], simple CNN models extract features from a time series of both modalities, concurrently leveraging spatial and temporal contexts. In [33] the authors investigate domain adaptation techniques applied to various machine learning algorithms. Notably, [3, 58, 92] focus on post-flood imagery and experiment with CNN architectures distinguishing flood water from permanent water, assisted by auxiliary DEM input. Transfer learning approaches between SAR and multispectral domains have been explored in works like [46], [39], and [26], whereas in [31] a U-Net-like architecture with transformer modules is trained on Sentinel-2 pre-flood and Sentinel-1 post-flood images.

Nevertheless, multiple studies have focused solely on the use of SAR imagery due to its all weather imaging capabilities, and have designed various CNN architectures in order to segment the post-flood SAR image into flooded/non-flooded pixels (e.g. [38], [60], [85], [86], [13], [35], [1]). On the other hand, a number of methods opt for bitemporal imagery as input to their models in order to perform

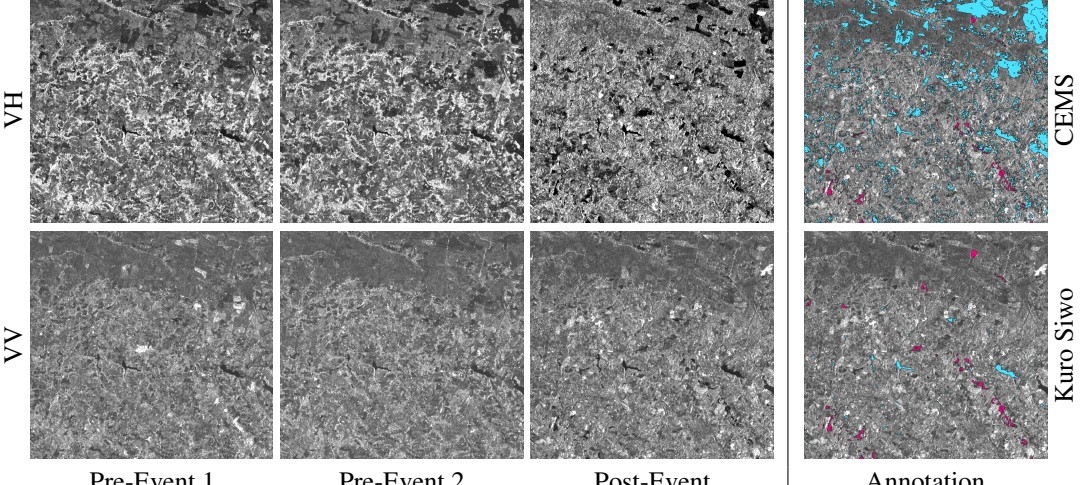

Figure 2: (left) Mosaic depicting Kuro Siwo samples for both VV and VH polarizations. (right) Copernicus Emergency Management Service (CEMS) annotations for a 2020 flood event in an agricultural area in France vis à vis Kuro Siwo photointerpretation. Cyan denotes permanent water bodies while purple indicates flooded areas. Notably, errors in CEMS annotations are apparent, particularly in the permanent waters class, suggesting the possibility that the CEMS annotator solely relied on VH polarization for annotation in this particular example. Quantitatively, the two products exhibit IoU of 51% and 48% for the permanent water and flood categories respectively.

change detection analysis and better isolate the flooding events. For example, [34] utilize simple dual-branch CNN architectures on bitemporal SAR data, whereas [20], [90] and [71] exploit transformer networks for better feature and context extraction. In [93] a dual-branch CNN is pretrained on water extraction and then finetuned on flood inundation mapping, while in [27] a sparse autoencoder is employed for the creation of pseudo-labels which are then used to train a simple small CNN. Finally, in [84] the model is also assisted by salient maps produced using the backscatter coefficient, and in [88] a time series of pre-flood SAR acquisitions is fed as input to a U-Net with ConvLSTM modules which is trained in a contrastive self-supervised way.

## 3   Kuro Siwo dataset

**Requirements:** While substantial efforts have been invested in automated flood mapping, the absence of a diverse and well-curated remote sensing dataset hampers the potential of deep learning methods for this critical task. To address this gap, we present Kuro Siwo, a distinctive flood inundation mapping dataset constructed with specific constraints. Firstly, the dataset is SAR-based, ensuring availability of usable imagery in all weather conditions, day and night, even during production in an operational context. In addition, we stick to SAR imagery from the Sentinel-1A&B satellites, since Copernicus data are available on a free and open basis, and there is strong commitment for the continuity of the mission. Secondly, Kuro Siwo is designed as a multi-temporal dataset to enable distinguishing between permanent water bodies, e.g rivers and lakes, and flooded areas, and also to facilitate testing change detection computer vision architectures. Thirdly, Kuro Siwo aims for diversity, featuring extensive spatio-temporal coverage of 43 major flood events spanning from 2015 to 2022, across six continents and three climate zones. The omission of flood events in Antarctica and polar/cold climate zones reflects their real-life distribution, as these areas typically do not experience floods. Fig. 1 illustrates the spatial coverage of Kuro Siwo, including the distribution of climate zones. Lastly, we prioritize the generation of high-quality flood annotations at a global scale, achieved through manual photointerpretation by a group of SAR experts, surpassing those from existing sources like the Copernicus Emergency Management Service (CEMS).

**Input data:** For each event, we assemble a triplet of Sentinel-1 data at two forms: a) Level-1 GRD SAR data and b) Level-1 SLC SAR data, both at 10 m spatial resolution. This triplet comprises two pre-event images with varying temporal distances — eliminating rigid constraints for real-world applications — and one post-event image acquired as close as possible to the actual event date. The

temporal gap depends on the sensor revisit time and the location on earth, resulting in a mean of 3.6 days (with std of 6.07 days) and a median of 1 day in post-event captions. We impose that for each flood event, all three SAR images belong to either the descending or ascending imaging geometry to prevent variations in layover, foreshortening and shadow effects [16] within the same sample. Additionally, for each SAR image in Kuro Siwo, both VV (Vertical transmit and Vertical receive) and VH (Vertical transmit and Horizontal receive) polarizations are registered, as studies have demonstrated their complementary value for flood mapping [45, 32].

**GRD Preprocessing:** To prepare our data for deep learning methods, we employ a standard Sentinel-1 GRD preprocessing pipeline [25] through the Sentinel Application Platform (SNAP) [95]. This pipeline involves precise orbit application, removal of thermal and border noise, land and sea masking, calibration, speckle filtering and terrain correction using an external digital elevation model (DEM), i.e. SRTM 1 Sec. The output of this pipeline yields a radiometrically calibrated SAR backscatter.

**SLC preprocessing:** Furthermore, we develop a minimal preprocessing pipeline for the SAR SLC data with two goals: a) create a foundation dataset of unrefined georeferenced SLC SAR data paired with quality annotations, allowing the end user to make all subsequent processing decisions, and b) provide the first baseline models utilizing complex valued input with both the phase and amplitude information on Kuro Siwo. Our processing pipeline includes swath selection, precise orbit application and debursting. We then extract the phase and amplitude information from the complex data and apply terrain correction using an external DEM (SRTM 1 Sec).

**Dataset creation:** The events included in Kuro Siwo can be split in two categories: a) events that have been previously annotated by the CEMS and b) events without any publicly available annotation. When CEMS annotations are available, we initialize the labeling process by utilizing the original CEMS shapefiles; otherwise we begin from scratch. Subsequently, we engage, with a team of five SAR experts, in the photointerpretation of the GRD preprocessed images and generate the ground truth masks assigning each pixel to one of three categories, i.e. Permanent Waters, Floods and No Water. The detailed annotation procedure and principles can be found in the Supplemental Material. The resulting flood mapping dataset contains $67,490$ timeseries with $202,470$ unique SAR samples stored as $224 \times 224$ tiles, along with all necessary metadata including the caption dates, the respective climate zone, the id of the area of interest, the Digital Elevation Model (obtained from SRTM 1Sec) etc. We provide both GRD and the SLC processed products for all events, along with deep learning ready time series-reference maps pairs. Kuro Siwo is released under the MIT License.[3]

**Why update CEMS annotations?** Some existing datasets used for rapid flood mapping with Sentinel-1 SAR data rely on the freely available CEMS annotations, which cover a large number of events globally. Typically, these annotations involve a combined thresholding and photointerpretation approach during a CEMS Activation. There are three significant challenges associated with CEMS annotations. Firstly, these Activations require the delivery of flood delineation products within an exceptionally short time frame, usually a few hours upon receiving the satellite imagery, leading to human errors in the annotations. Secondly, different teams within Copernicus annotate various flood events, resulting in variable photointerpretation and methodological biases in the annotations. Lastly, quality standards requirements have evolved in CEMS over the years, and consequently, the latest flood activations provide better quality annotations compared to older ones.

The above result in problematic annotations that hamper the potential to train robust and accurate deep learning models. For example, the creators of the Ombria [22] and the MMFlood [56] datasets (Tab. 1) that both use CEMS annotations, report low classification accuracies. This was our experience also when we used the original CEMS annotations; our deep learning models achieved less than 70% for all evaluation metrics. Therefore we decided to invest in updating CEMS annotations through photointerpretation. Fig. 2 exemplifies the annotation improvement attained in Kuro Siwo for a flood event in an agricultural area in France. The errors in CEMS annotations, especially for the permanent waters class, are obvious when carefully examining both polarizations. To discern the disparity between the CEMS and Kuro Siwo annotation for the specific flood event, the two products exhibit Intersection over Union (IoU) of $51\%$ and $48\%$ for the permanent water and flood categories, respectively.

**Going beyond CEMS:** Building on CEMS annotations, which focuses primarily on Europe, results in severe underrepresentation of other continents. Recognizing this, we expand our study on flood

---

[3]https://opensource.org/license/MIT

events from Asia, Australia, Africa as well as South and North America, aiming for a more balanced spatial distribution. These events cover extensive areas providing a substantial number of additional training patches. Floods in Europe are fragmented small scale events, as opposed to large flood events encountered in other continents, e.g. Asia. Consequently, while most events in Kuro Siwo are from Europe compared to Asia, the latter contributes with more samples.

**Dataset split:** Scholastically assessing and comparing the capacity of flood mapping models to operate in novel environments is of great importance to reliably evaluate future methods and eventually deploy them in the real world. With that in mind, we construct a challenging evaluation framework, selecting 10 flood events across the globe as testing sites, covering a wide range of environmental conditions representing all six continents and three major climate zones featured in Kuro Siwo. The spatial distribution of training, validation and test flood events in Kuro Siwo is illustrated in Fig. 1 along with the continents representation.

**Unlabeled component:** In the context of natural hazards and extreme events, instances of positive occurrences, such as floods, volcanic unrest [9], landslides [7] and wildfires [41], are notably rare. This scarcity results in a limited dataset, insufficient for mapping the diverse, intricate and dynamic environmental variables, including water. Furthermore, the number of flash floods monitored with publicly available SAR imagery is finite, given that Sentinel-1 data are available after 2014 only, while at the same time, acquiring and photointerpreting them all implies a substantial cost. Recognizing these constraints and the importance of generalizing to unseen events, we offer an extensive, unlabeled collection of satellite frame triplets, adhering to the same principles and preprocessing pipeline as the annotated Kuro Siwo set. This resultant dataset encompasses $533,847$ time series with $1,601,511$ unique SAR samples. Motivated by recent advancements in foundational models for computer vision [40], we release this dataset to encourage the exploration of SSL methods specifically designed for the domain of rapid flood mapping.

## 4 BlackBench: An extensive benchmark on rapid flood mapping

Building on Kuro Siwo, we introduce BlackBench, an extensive benchmark employing a large set of powerful models under various configurations to act as strong baselines for future methods. BlackBench includes common semantic segmentation architectures like U-Net [69], DeepLabv3 [12] and UPerNet [87] using both convolutional and transformer backbones like the ResNet [30] family, Swin Transformer [47] and ConvNext [48]. We use two variants of each backbone for UPerNet indicated as Small (S) and Base (B) as defined in [15]. Furthermore, we include a set of models inspired by change detection problems, like FC-EF-Diff and FC-EF-Conc [17], which were among the first deep learning models proposed for this task. Additionally, we include SNUNet-CD [24], a densely connected U-Net++ [94] model with a dual-branch encoder, and Changeformer [4], a model consisting of two siamese branches with transformer blocks and a lightweight MLP decoder. Finally, we explore ConvLSTM [73], a recurrent convolutional architecture suitable for time-series data. We employ an encoder-decoder architecture, and we select only the last output map for an N-to-1 scheme.

For each model included in BlackBench, we assess the performance across diverse input scenarios, using GRD data. The input setting varies on two factors. First, in regard to the time series length and second, in regard to the inclusion of elevation information. For all semantic segmentation models we consider time series of either 2 or 3 images varying the number of pre-event captions. For the change detection models we strictly use two captions by selecting only the most recent pre-flood image since this family of architectures typically employs a two-stream input encoder. For each time series length, we examine the performance of the models when we a) include a DEM, b) include slope information (derived from the DEM) and c) provide no elevation information at all. In Tab. 2 we report the performance of the best setting for each architecture. In particular, we report the F1-Score (F1) for each class as well as the overall mean IoU (mIoU). Additionally, we evaluate the F1-Score for the binary task of water detection by combining the predictions for permanent water and flood into one "water" class. All semantic segmentation models use pretrained backbones and were trained for 20 epochs. U-Net and DeepLabv3 models use backbones pretrained on ImageNet [18], while UPerNet uses backbones from [15]. Change detection and recurrent models were trained from scratch for 150 epochs. Finally, we create a baseline model trained via SSL using the unlabeled component of Kuro Siwo, excluding the events from the test set. To model interactions in the temporal dimension as well as make the finetuning process more direct, we treat each Kuro Siwo triplet as one sample. We utilize the Masked Autoencoder [29] (MAE) method employing a vision transformer [21] (ViT) with 24

Table 2: This table presents the best performing setting for each architecture utilising the GRD data, in regards to the time series length as well as the utilization of elevation information. "No water", "permanent water", "flood" and "water" classes are represented by NW, PW, F and W respectively. Best values are marked in **bold**, second best are underlined.

| Model | Caps. | DEM | Slope | F1-NW(%) | F1-PW(%) | F1-F(%) | mIOU(%) | F1-W(%) |
|---|---|---|---|---|---|---|---|---|
| UNet-ResNet18 | 2 | - | - | 98.72 | 76.01 | 79.86 | 75.12 | 83.31 |
| UNet-ResNet50 | 3 | - | - | 98.73 | 78.24 | **80.12** | **76.20** | **83.85** |
| UNet-ResNet101 | 3 | - | ✓ | 98.69 | **79.33** | 78.92 | 76.12 | 82.88 |
| DeepLab-ResNet18 | 3 | - | ✓ | **98.74** | 77.35 | 78.51 | 75.07 | 83.38 |
| DeepLab-ResNet50 | 3 | - | ✓ | 98.71 | 77.35 | 78.70 | 75.14 | 83.20 |
| DeepLab-ResNet101 | 2 | - | - | 98.65 | 77.03 | 78.46 | 74.84 | 82.52 |
| UPerNet-SwinS | 2 | - | ✓ | 98.70 | 77.59 | 78.95 | 75.35 | 82.51 |
| UPerNet-SwinB | 3 | - | - | 98.72 | 76.94 | 79.28 | 75.22 | 83.15 |
| UPerNet-ConvnextS | 3 | - | - | 98.65 | 77.09 | 79.3 | 75.26 | 82.86 |
| UPerNet-ConvnextB | 2 | - | - | 98.60 | 77.00 | 78.15 | 74.66 | 82.57 |
| FloodViT | 3 | - | - | 98.73 | 75.23 | 78.70 | 74.22 | 83.21 |
| FloodViT-FT | 3 | - | - | **98.74** | 75.45 | 78.35 | 74.17 | 83.03 |
| FC-EF-Diff | 2 | ✓ | - | 97.8 | 10.93 | 74.27 | 53.52 | 62.6 |
| FC-EF-Conc | 2 | ✓ | - | 98.64 | 71.89 | 75.06 | 71.17 | 82.04 |
| SNUNet-CD | 2 | - | - | 98.57 | 73.14 | 74.26 | 71.3 | 81.51 |
| Changeformer | 2 | - | - | 98.66 | 74.84 | 76.96 | 73.23 | 82.23 |
| ConvLSTM | 3 | - | - | 98.66 | 76.57 | 77.53 | 74.23 | 81.96 |

layers and 16 attention heads as our encoder and train for 100 epochs. We stack the triplets on the channel dimension as done with all segmentation methods in BlackBench. To model the varying temporal distances between the pre-event and the post-event captions, we include three temporal embeddings, i.e. one for each timestep. We encode the year, month and day independently using a sinusoidal encoding and concatenate them. The resulting temporal embeddings are added to the tokens as done with the standard, learnable positional embeddings. For the segmentation downstream task, we add a simple, trainable, convolutional decoder on top of the learnt representations. We evaluate the capacity of our model by a) keeping the encoder frozen and training only the decoder module, and b) finetune the last 12 transformer layers along with the decoder. We will refer to these models as FloodViT and FloodViT-FT respectively. We discuss the insights gained from BlackBench in detail in Sec. 5. Furthermore, we provide BlackBench results for the SLC component in the Supplementary material.

## 5 Discussion

**Kuro Siwo evaluation:** Quantitative comparison of Kuro Siwo with other datasets is challenging, as most publicly available SAR-based datasets do not follow the same evaluation principles. These principles include testing on events absent from the training set and ensuring a diverse array of test events.

We identify MMFlood as the most fitting dataset for our comparison, offering 34 GRD test events with labels acquired from CEMS. The authors report a maximum of 66.52% mIoU solving the binary flood/no flood task, which is significantly lower than the one presented in BlackBench, where we solve a multi-class problem. This may be an indirect consequence of the problematic nature of CEMS

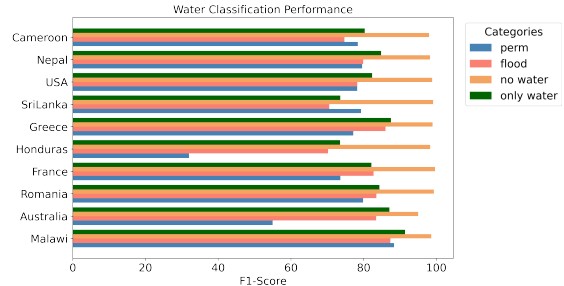

Figure 3: Per-event performance of the best model in BlackBench on the test set.

labels as discussed in Sec. 3 and an indicator for the importance of accurate and cross verified ground truth labels. Another likely reason could be the oversight of the temporal dimension, as only post-event images are being used. Examining Tab. 2, we observe that the temporal aspect is crucial for semantic segmentation models, with most performing best when utilizing all available pre-event images. Conversely, incorporating DEM information offers negligible improvements. Even when

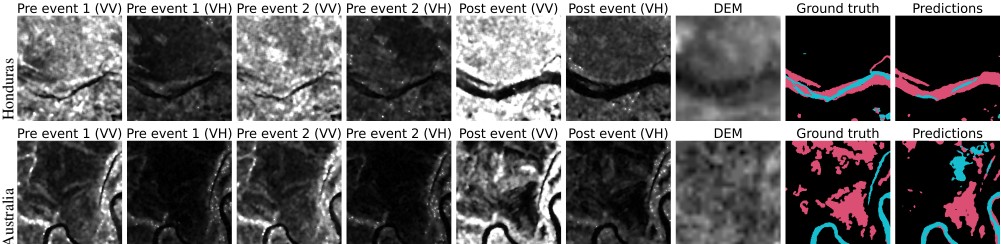

Figure 4: Qualitative evaluation of our best model in Honduras and Australia. Flood is marked in purple, permanent waters in cyan and non-water areas in **black**.

including DEM yields the best results for a model, the performance gains are marginal. Interestingly, we notice a consistent discrepancy across models in Tab. 2 between the reported metrics for permanent water bodies and the rest of the classes. We hypothesize that this behaviour stems from the inherent challenge of discerning permanent water bodies from flooded areas in specific locations, especially close to overflowed rivers and lakes.

This problem is magnified by the constant flow of water, resulting in small changes in the area it populates which in turn materializes with different signatures for each SAR timestep. This effect has a stronger presence in Fig. 3, where we present the per-event performance of the best model in BlackBench, i.e. UNet-ResNet50. Overall, we notice a stable performance despite the variable environmental conditions between events. However, performance in permanent waters detection drastically decreases for Honduras and Australia. We provide a qualitative examination of samples from these areas in Fig. 4. Indeed, we observe good detection of both flooded areas and permanent water bodies, however it is particularly challenging to accurately delineate their boundaries when they overlap.

Notably, FloodViT performs comparably with most models and surpasses all change detection methods despite having only a small decoder trained for the segmentation task. We did not observe additional benefits from fine-tuning FloodViT-FT. Further investigation on SSL methods for this task is left as future work.

**SLC dataset:** The inclusion of the unrefined SLC products paired with quality annotations at a global scale is an important aspect of Kuro Siwo. This component frees the end user of the task-specific preprocessing choices of the dataset creators, while enabling the investigation of important research questions such as the exploitation of the complex valued signal nature of SAR data. Our initial experiments (see Tab.2 in Supplementary material) suggest that DL models trained on SLC data are capable classifiers even on such unrefined data. In fact, a simple UNet with a ResNet18 backbone is able to achieve $\approx 79.94\%$ F-Score on the binary water detection task and $\approx 71.20\%$ and $\approx 76.76\%$ on the flood and permanent waters categories respectively. This is particularly promising since models in BlackBench are not tailored to the unique characteristics of SLC data. The thorough exploration of the complex information of SAR data is left as future work.

**Limitations:** Despite significant efforts to create a quality and diverse flood mapping dataset, approximately half of Kuro Siwo's flood events are from Europe, with the rest distributed globally, indicating a need for more balanced spatial coverage. Furthermore, while Kuro Siwo can train models to map observed flood extents from SAR images, accurately capturing the actual maximum flood extent remains dependent on the revisit time of the Sentinel-1 mission. Finally, the inherent challenges associated with SAR data are a key limitation of Kuro Siwo. For example, speckle effect introduces granular noise that complicates the differentiation between flooded and non-flooded areas. Complex terrain and varying land cover types, such as forests, further complicate flood detection by affecting radar signal interaction. Variations in water surface roughness due to wind or the existence of vegetation can alter backscatter, making it difficult to consistently identify floodwaters. Detecting floods beneath dense vegetation is also challenging, as the S1 C-band radar signal may not penetrate sufficiently (as opposed to L-band SAR sensors) to reveal underlying water. Urban areas pose another difficulty, as radar signals often produce complex scattering effects (e.g. double-bounce reflections), making it difficult to distinguish between water and other surfaces, like wet roads or buildings.

**Future research directions:** Multispectral data, such as those provided by the Sentinel-2 constellation, can significantly aid flood mapping under cloud-free conditions since they provide a clearer view of the underlying waters compared to SAR. Incorporating multispectral imagery in Kuro Siwo

would require extensive photointerpretation, as acquisition timestamps between the satellites would possibly differ leading to discrepancies in detected flood events, given the short life cycle of flash floods. Nevertheless, the synergistic use of both datasets is a promising research direction, combining the precision of multispectral data with the all-weather resilience of SAR. Having Kuro Siwo as a publicly available resource with reliable SAR based annotations can provide the foundation for assembling such a multi-modal dataset.

Moreover, including a permanent water layer could be beneficial for discriminating between the two water classes. However, water bodies exhibit dynamic variable extents due to a variety of factors including meteorological conditions and environmental changes. Using a static water layer as input to the models could potentially impute noise and ultimately disrupt the learning process. Refining existing layers for each timestep in Kuro Siwo could prove counterintuitive in production as rapid flood mapping is of utmost importance. Further investigation on such auxiliary information is left as future work.

Finally, a related but fundamentally different topic than rapid flood mapping is flood forecasting. Kuro Siwo is focused on supporting emergency response activities within short time frames rather than long-term monitoring of water bodies for disaster resilience. While crucial, this task demands a distinct approach and additional data, such as local weather patterns, water storage capacity, soil data, and denser time series. Future research on this topic could strongly complement Kuro Siwo.

## 6    Conclusion

In this work, we release Kuro Siwo, a global, multi-temporal SAR dataset for flood mapping that includes both Level-1 GRD and SLC products. Kuro Siwo features manual annotations for 43 flood events worldwide, providing time series SAR data with dual polarization and elevation information. Alongside the curated dataset, we also release a large unlabeled dataset for large-scale self-supervised learning. Additionally, we introduce BlackBench, the first unified benchmark on Kuro Siwo, offering strong baselines for rapid flood mapping. We strongly believe that the release of Kuro Siwo will propel research in the crucial task of rapid flood mapping, potentially aiding in disaster response and relief management.

## Acknowledgments and Disclosure of Funding

This work has received funding from the project ThinkingEarth (grant agreement No 101130544) and from the project MeDiTwin (grant agreement No 101159723) of the European Union's Horizon Europe research and innovation programme.

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
