# Kuro Siwo: 33 billion $m^2$ under the water
# A global multi-temporal satellite dataset for rapid flood mapping

# Supplemental material

## 1 Dataset

The total size of the compressed dataset is $\approx 1.33$ TB, with the GRD component (including the DEMs and metadata) taking $\approx 705.8$ GB and the SLC $\approx 492.6$ GB.

All code and data will be maintained at the project's repo.

## 2 Evaluation on a new, unseen event

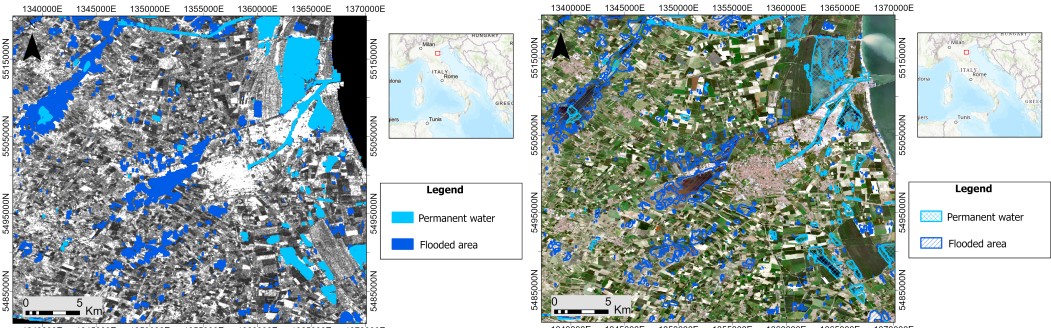

Figure 1: Predictions of the flood and permanent water extent using our best model, i.e. Unet-ResNet50, on Emilia-Romagna floods in May 2023. On the left hand side we present the post-event SAR image used for the prediction, captured in 22/05/2023, while on the right hand side the respective Sentinel-2 RGB image captured in 23/05/2023 (one day later).

In Fig. 1 we assess the performance of our best model, i.e. Unet-ResNet50, on the recent floods of Emiglia-Romana, Italy, which took place on May 2023. Our predictions were based on a post-event SAR image acquired on 22/05/2023, and two pre-event SAR images from 10/05/2023 and 28/04/2023. On the right hand side of Fig. 1 we overlay our predictions with a Sentinel-2 RGB image captured on 23/05/2023 for better inspection of the annotated areas. Examining both derived maps we clearly see the capacity of the proposed model, showcasing the quality of Kuro Siwo. Our model manages to uncover all major water bodies, such as the rivers and ponds in the Pialassa della Baiona, as well as flooded regions of diverse sizes. Given that BlackBench presents baseline models, the performance of future task-specific methods may potentially surpass the baselines offered in this work.

## 3 Additional results

Fig. 2 showcases representative samples from the Kuro Siwo GRD dataset, providing a closer examination of flood events spanning a diverse set of countries. The predictions come from the best performing model in BlackBench, i.e UNet-ResNet50. The results underscore the significance of

leveraging both polarization bands (VV and VH) for an accurate evaluation of the influence of the underlying land cover. While the VV band is effective in detecting the majority of flooded areas, the post-flood VV imagery may also include some bright regions in flooded areas that might be mistaken for dry land. These areas are refined and clarified by incorporating the VH band, which exhibits a more uniform dark region with smaller bright patches. For example, the flooded area in Fig. 2e is more concretely depicted in the VH band with stronger backscatter in the regions farthest from the river which also present a challenge to our model.

On the contrary, Fig. 2b, Fig. 2g and Fig. 2e feature specific dark regions in the pre-flood imagery that are not labeled as water bodies in the annotation. These areas may correspond to isolated pixels experiencing the speckle effects or to crop parcels or vegetated areas with varying moisture content, that are in different phenological states compared to neighboring regions. In general, the model manages to successfully capture the outline of most water elements, albeit often missing narrow rivers, which is a rather difficult task considering the dynamic nature of water flow and the changes of river width over time.

Furthermore, Fig. 2c and Fig. 2d present instances of river overflow. The Kuro Siwo annotation precisely delineates the initial river shape as inferred by the two pre-event captions, along with the flooded river banks marked in purple. In Fig. 2d, the Unet-ResNet50 achieves high accuracy in both permanent waters and flood, whereas in Fig. 2c it seems to miss the exact shape of the river in which is depicted in a more solid black region in the VH band. Furthermore, the overflowed river banks present a significant challenge for the model which is constantly confused between the two water classes.

Finally, Fig. 2h presents a particularly challenging sample, wherein the small reservoir of permanent water in the upper-left of the patch is difficult to detect due to the strong presence of SAR speckle noise.

# 4   Annotation process

The outlined annotation process unfolds in distinct phases: initially, we establish the entire procedure by defining the annotation categories, i.e. "Permanent Waters", "Floods" and "No Water". We extract the corresponding photointerpretation keys, set the annotation scale to 1:1000, and establish criteria for flagging artifacts in the input data. Next, we partition the 43 flood events into five segments. A team of five SAR experts undertook the annotation process, with each member assigned to annotate one part and quality-check the annotations of another team member. The supervision of this process was overseen by a senior remote sensing scientist. Regular group meetings were conducted to address and resolve any potential discrepancies between the primary annotator and the responsible for the quality check.

# 5   Training setup

In Table 2 we report the training hyperparameters for each model in BlackBench. All SAR images were clipped at a max value of 0.15 and normalized to 0 mean and 1 standard deviation. At training time we skip samples that do not contain water (regardless of the class). We test on all available samples of the test areas. All models utilize the cross-entropy loss function, apart from SNUNet-CD that employs a compound loss consisting of cross-entropy and Dice loss, in line with the original publication.

# 6   SLC benchmark

In Table 1 we present the BlackBench results for the SLC component of Kuro Siwo.

# 7   Computational resources

All experiments were conducted on a single NVIDIA GeForce RTX 3090 Ti.

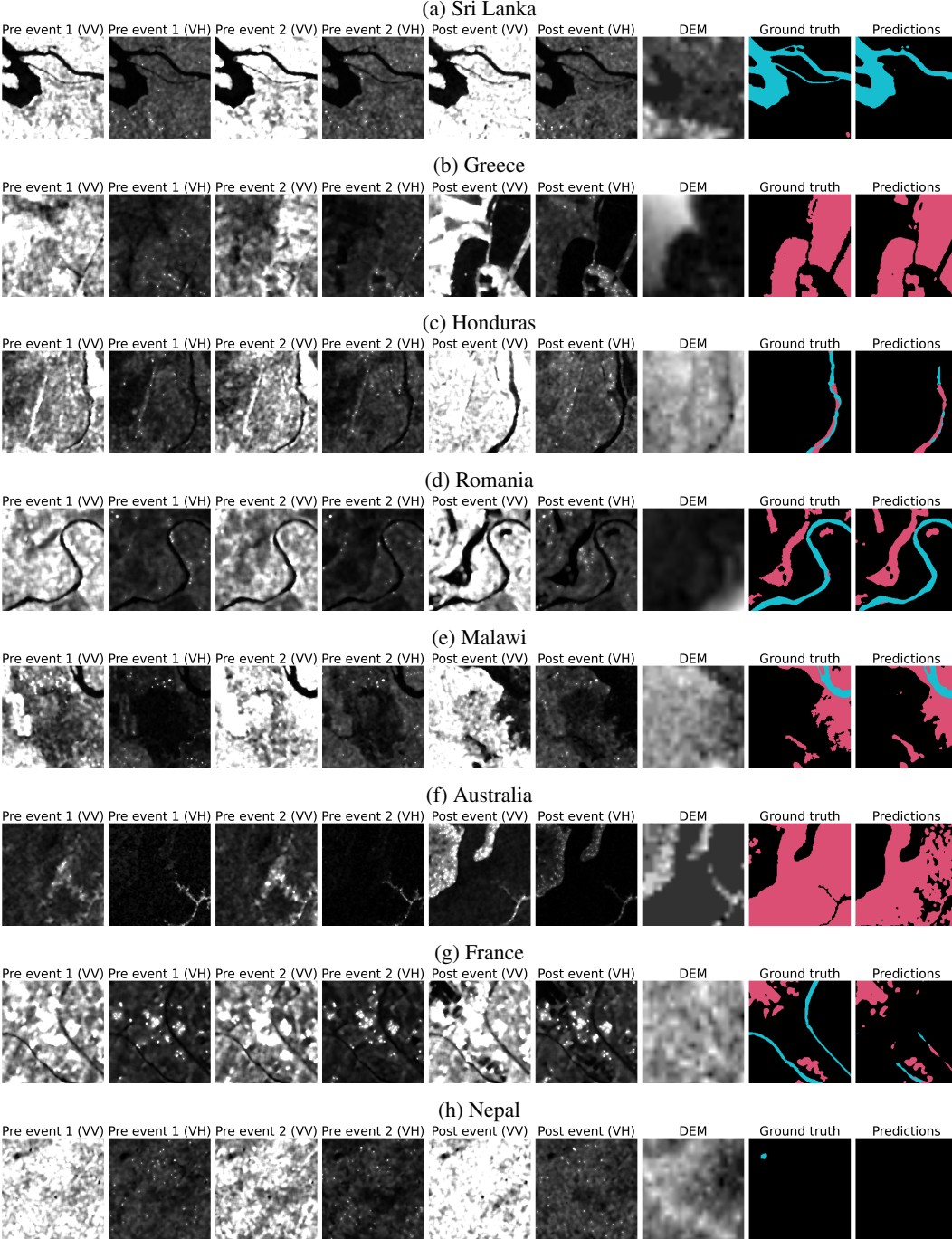

Figure 2: Kuro Siwo GRD samples. Flood is marked in purple, permanent waters in cyan and non-water areas in **black**. The predictions come from the best performing model in BlackBench, i.e UNet-ResNet50.

# 8   Intended uses

Kuro Siwo is designed to address two primary objectives: a) overcoming the lack of large, meticulously annotated datasets for flash flood mapping and b) promoting the development of deep learning methods that utilize both phase and amplitude information, without restrictive preprocessing choices. The intended uses of Kuro Siwo are thus, a) the development of robust and accurate flash flood

Table 1: This table presents the best performing setting for each architecture for the SLC component, in regards to the time series length as well as the utilization of elevation information. "No water", "permanent water", "flood" and "water" classes are represented by NW, PW, F and W respectively. Best values are marked in **bold**, second best are underlined.

| Model | Caps. | DEM | Slope | F1-NW(%) | F1-PW(%) | F1-F(%) | mIOU(%) | F1-W(%) |
|---|---|---|---|---|---|---|---|---|
| UNet-ResNet18 | 3 | - | ✓ | **97.88** | 76.76 | **71.20** | 71.14 | **79.94** |
| UNet-ResNet50 | 3 | - | ✓ | 97.68 | 78.76 | 68.91 | 71.00 | 78.10 |
| UNet-ResNet101 | 3 | - | ✓ | 97.68 | 76.64 | 69.74 | 70.38 | 77.81 |
| DeepLab-ResNet18 | 3 | - | ✓ | 97.63 | 74.21 | 69.15 | 69.08 | 78.51 |
| DeepLab-ResNet50 | 2 | - | ✓ | 97.63 | 76.57 | 66.20 | 68.96 | 77.10 |
| DeepLab-ResNet101 | 3 | - | - | 97.63 | 74.92 | 69.72 | 69.60 | 77.13 |
| UPerNet-SwinS | 3 | - | - | 97.71 | **79.48** | 70.39 | **71.93** | 78.17 |
| UPerNet-SwinB | 3 | - | - | 97.63 | 79.16 | 70.25 | 71.68 | 77.23 |
| UPerNet-ConvnextS | 3 | - | - | 97.64 | 78.93 | 69.33 | 71.22 | 77.45 |
| UPerNet-ConvnextB | 3 | - | - | 97.71 | 78.79 | 69.70 | 71.34 | 77.51 |
| FloodViT | 3 | - | - | 97.32 | 60.03 | 56.87 | 59.13 | 69.76 |
| FloodViT-FT | 3 | - | - | 96.84 | 51.23 | 57.62 | 56.26 | 61.43 |
| FC-EF-Diff | 2 | ✓ | - | 97.53 | 70.51 | 69.13 | 67.49 | 75.97 |
| FC-EF-Conc | 2 | - | - | 97.72 | 71.92 | 68.78 | 68.03 | 77.3 |
| SNUNet-CD | 2 | - | ✓ | 97.65 | 75.31 | 68.82 | 69.42 | 77.99 |
| Changeformer | 2 | ✓ | - | 97.61 | 69.31 | 69.34 | 67.14 | 75.87 |
| ConvLSTM | 3 | - | - | 97.76 | 78.16 | 70.58 | 71.44 | 78.0 |

Table 2: The training hyperparameters used in BlackBench. LR stands for learning rate and SGD for Stochastic Gradient Descent.

| Model | Optimizer | LR | LR scheduling | Weight decay | # epochs |
|---|---|---|---|---|---|
| UNet-ResNet18 | Adam | 1e-3 | Cosine | - | 20 |
| UNet-ResNet50 | Adam | 1e-3 | Cosine | - | 20 |
| UNet-ResNet101 | Adam | 1e-3 | Cosine | - | 20 |
| DeepLab-ResNet18 | Adam | 1e-3 | Cosine | - | 20 |
| DeepLab-ResNet50 | Adam | 1e-3 | Cosine | - | 20 |
| DeepLab-ResNet101 | Adam | 1e-3 | Cosine | - | 20 |
| UPerNet-SwinS | Adam | 1e-4 | - | 1e-4 | 20 |
| UPerNet-SwinB | Adam | 1e-4 | - | 1e-4 | 20 |
| UPerNet-ConvnextS | Adam | 1e-4 | - | 1e-4 | 20 |
| UPerNet-ConvnextB | Adam | 1e-4 | - | 1e-4 | 20 |
| FloodViT | Adam | 1e-3 | Cosine | 1e-5 | 20 |
| FloodViT-FT | Adam | 1e-4 | Cosine | 1e-3 | 20 |
| FC-EF-Diff | Adam | 1e-5 | - | - | 150 |
| FC-EF-Conc | Adam | 1e-5 | - | - | 150 |
| SNUNet-CD | Adam | 1e-5 | - | - | 150 |
| Changeformer | SGD | 6e-4 | Linear | 1e-5 | 150 |
| ConvLSTM | Adam | 1e-3 | - | - | 150 |

mapping methods, which could potentially be applied in operational settings, and b) research in the direct exploitation of complex SAR data.

# 9 License

The authors state that they bear all responsibility in case of any rights violation. The Kuro Siwo dataset as well as BlackBench are released under the MIT License.[1]

---

[1]https://opensource.org/license/MIT

## 10 Maintenance plan

The main hosting platform of Kuro Siwo and BlackBench is GitHub and the authors guarantee access to the main code, as well as maintenance and issue tracking. Due to their seer size, data are hosted at a major cloud storage provider, ensuring seamless availability and adequate download speeds.