# OpenReview forum: "Kuro Siwo: 33 billion $m^2$ under the water. A global multi-temporal satellite dataset for rapid flood mapping"
_NeurIPS.cc/2024/Datasets_and_Benchmarks_Track — NeurIPS 2024 Track Datasets and Benchmarks Poster_

### Official Review · Reviewer_nnn6 · 2024-07-22
**Paper is well written and the workload is sufficient. However, the superiority compared to previous datasets may be not enough such as the small collected event number.**

**Rating:** 6
**Confidence:** 4
**Clarity:** The paper is well written.

**Review:**

1 The most superiority compared to previous datasets lies that the authors provide both and coregistered GRD and SLC SAR data, while its necessity and performance is not presented.
2 The number of collected flood event is relatively small as in Table 1.
3 Please detail the labeled sample number and unlabeled number in Table 1.
4 Although time-series data are provided, you use the time-steps in the manner the same as the single-step dataset. How do you utilize the time-series information?
5 The number of samples seems being large by including the unlabeled and time-series images. The images of labeled flood event may be not large and the comparison in Table 1 may be not fair.

**Strengths:**

1 Both and coregistered GRD and SLC SAR data.
2 Manual and more accurate annotation.
3 The paper is well written.

**Additional Feedback:**

None.

**Correctness:**

Yes, the claims made in the submission are correct, and the dataset is constructed in a sound way.

**Documentation:**

Yes, there is sufficient detail on data collection and organization, availability and maintenance, and ethical and responsible use.

**Ethics:**

No, I don’t suspect there are any ethical concerns with the submission that warrant further discussion or review.

**Limitations:**

No, the superiority compared to previous datasets is not well clarified.

**Opportunities For Improvement:**

More fair comparison and the clarify of the dataset superiority is needed.

**Relation To Prior Work:**

The most superiority compared to previous datasets lies that the authors provide both and coregistered GRD and SLC SAR data.

**Summary And Contributions:**

Summary: The paper introduces a global, multi-temporal SAR dataset for flood mapping that 353 includes both Level-1 GRD and SLC products.

---

> ### Author Rebuttal · Authors · 2024-08-17
>
> We thank the reviewer for the thoughtful review.
>
> Necessity of SLC data:
>
> The GRD component, while optimised for flash flood mapping, sacrifices phase information— phase statistics, signal decorrelation and coherence maps can be used in tasks like change detection, particularly being a good driver for flood segmentation. As discussed in L190-196, the introduction of the SLC data provides both amplitude and phase information enabling the investigation of future methods that exploit this complex valued input.
>
> Given the unlabelled component of Kuro Siwo, the potential for representation learning is greatly expanded, opening up a wide array of applications. By providing SLC data with minimal preprocessing, we offer users the flexibility to tailor Kuro Siwo to their specific downstream tasks by applying their own preprocessing pipeline, without affecting the integrity of the dataset. This approach frees up pre-processing restrictions applicable for the GRD-based pipeline, and enhances the dataset's versatility.
>
> Additionally, we recognise that including benchmarks for models utilising SLC data would be valuable for future works that aim to create methods tailored for SLC data, hence we provide benchmark results (Preliminary results can be found in Table 2 of the uploaded pdf file). Our top-performing SLC model, UPerNet with a Swin-S backbone, achieves an F1-Score of 70.39% for the flood class, 79.48% for the permanent water class, and 78.17% for the binary water/no water task. Our second-best model, UPerNet with a Swin-B backbone, demonstrates comparable performance with an F1-Score of 77.23% on the binary task, 70.25% for the permanent water class, and 79.16% for the flood category. These results highlight the strong predictive skill of the BlackBench models in delineating floods from Level-1 SLC complex satellite products. This is the first time that such low-level complex-valued satellite products have been used to develop a user-ready final product, and we believe this paves the way for numerous new applications.
>
> Number of collected flood events:
>
> Kuro Siwo offers manual annotations for 43 flood events, making it one of the largest flood mapping datasets, with meticulously curated annotations. As seen in Table 1, only four other datasets surpass Kuro Siwo in terms of the number of events, but these rely on automatic, semi-automatic, or CEMS annotations. Kuro Siwo stands out as the largest Sentinel-1 flood mapping dataset with expert-level manual annotations. In fact, as can be seen from Table 1, the second largest flood dataset with manual annotations is CAU-Flood with 18 events, way less than Kuro Siwo (43).
>
> Labelled vs unlabelled samples:
>
> Table 1 shows the number of labelled samples in Kuro Siwo. Specifically, the dataset contains 67,490 labelled time series and 533,847 total time series (both labelled and unlabelled). This is described in L204 and L249. To improve clarity, we will add an extra row in Table 1 to differentiate between labelled and unlabelled components of Kuro Siwo more explicitly. We demonstrate that in Table 1 in the uploaded pdf file.
>
> Utilising time-series information:
>
> BlackBench explicitly models temporal information through various methods. Change detection models assume temporally ordered images, i.e. a pre-event and a post-event image, while FloodViT, as described in L283-288, leverages temporal embeddings for each timestep. These embeddings are generated by independently encoding the year, month, and day of the caption using a sinusoidal encoding and concatenating them. They are subsequently added to the tokens along with learnable positional embeddings before being fed to the transformer, thus imputing important temporal information to the learning process.
>
> In addition, we have now included ConvLSTM, a recurrent neural network that achieved promising results with 77.53% F1-Score for flood and 76.57% for permanent waters, in the camera-ready version (see Table 3 in the attached pdf). While we acknowledge there are multiple ways to model time series, our goal is not to provide a new SOTA for flood mapping but to present a quality dataset that fosters the development of such SOTA methods. BlackBench provides a strong benchmark and robust baselines to support this.
>
> Clarifications on dataset superiority:
>
> a ) As seen in Table 1 of the manuscript, Kuro Siwo i) is the largest publicly available SAR-based dataset (67,490 samples) containing ii) manually curated annotations (photo-interpretation by a team of experts) for flood mapping at a iii) global scale (43 events, 6 continents), along with SAR iv) time series (two before, one after the event) and elevation information. This holds the potential to propel research in this field as quality data are highly important for Deep Learning methods.
>
> b) Kuro Siwo addresses a significant gap in the literature, as discussed in Section 2 (L127-134), by providing the largest available source of SLC data paired with annotations. Unlike the GRD component, SLC data includes crucial phase information, which can enhance research on methods specifically designed for complex SLC data—a field that has been underexplored due to the lack of suitable datasets. Additionally, the inclusion of SLC data allows users to apply their own S1 preprocessing pipelines, circumventing the need to adopt the preprocessing decisions we made in Kuro Siwo for the GRD based pipeline. This flexibility extends to the unlabeled component as well, which supports representation learning and the development of encoders for various downstream tasks, accommodating different preprocessing requirements. This adds major flexibility to the dataset.
>
> c) The quality of the dataset is demonstrated with BlackBench, providing strong baselines capable of delineating flooded areas (see Table 2). Given a quality dataset, researchers can now focus on methodological improvements to further improve results.

---

### Official Review · Reviewer_vj9P · 2024-07-24

**Rating:** 5
**Confidence:** 3
**Correctness:** Strengths & Weaknesses
**Clarity:** Yes

**Review:**

Strengths & Weaknesses

**Strengths:**

1. The presented dataset contains massive manually created annotation by specialists, both filling in blanks and correcting errors in previously used datasets.

2. The benchmark created based on the presented dataset contains a diverse variety of state of the art semantic segmentation architecture.

3. The results exhibited reasonable improvement when the models are trained using the presented dataset.

**Additional Feedback:**

Strengths & Weaknesses

**Documentation:**

Yes

**Limitations:**

Yes

**Opportunities For Improvement:**

1. As the authors mentioned themselves in the paper, the flood events are heavily focused in Europe. Further more, they are also concentrated on coastal areas, while large rivers are also common flooding areas.

2. The labels of the waterbodies could have been further detailed with the help of the experts.

E.g., the a flooded waterbody can be a barrier lake or a oxbow lake and so on.

3. Despite flood mapping, the dataset also has a large potential in sequential prediction, which can be invaluable when it comes to forecasting the occurrence of flood and is long term impact. It would be great if the benchmark can also include such aspect.

**Relation To Prior Work:**

Yes

**Summary And Contributions:**

The paper presented a dataset which combined multi-polarization satellite images, elevation information along with manually annotated flood events worldwide. Massive unlabeled dataset is also provided to make up for the scarcity of such disastrous events. The authors also constructed a benchmark based on their dataset which contains a set of existing semantic segmentation models.

---

> ### Author Rebuttal · Authors · 2024-08-17
>
> We thank reviewer vj9P for their valuable feedback and constructive suggestions for improving our work.
>
> Spatial Coverage of the dataset:
>
> We acknowledge that our dataset contains a higher number of events from Europe, as discussed in L344. However, we made concerted efforts to mitigate this imbalance by including numerous events from underrepresented regions, as illustrated in Figure 1. Recognizing this inherent distribution challenge, we carefully constructed the train/validation/test split to evaluate the models' capacity to generalise to unseen locations (L234-240). Moreover, the absolute number of flood events does not necessarily reflect the total extent of flooded regions. For example, the total number of training patches in Asia (26,605) exceeds that in Europe (22,803). We took great care to include as many training samples as possible representing all 6 continents given the varying revisit rates of S1 across the different latitudes and the availability of post captions close to the event.
>
> Our comprehensive benchmark (Table 2), where we evaluated 16 architectures under various training setups (training >70 models), validates the ability of most models to generalise well across diverse environmental conditions, including non-European landscapes. While broader spatial coverage would undoubtedly be beneficial, we believe our dataset represents an important step forward for the research community.
>
> Flood events in coastal areas:
>
> It is important to clarify that the majority of the included events are not coastal. Approximately 38.67% of the Areas of Interest (AOIs) are in coastal regions.
>
> Refinement of waterbody labels:
>
> Waterbodies exhibit dynamic and variable extents due to a variety of factors, including meteorological conditions and environmental changes. Refining waterbody labels is therefore a non-trivial task that requires meticulous photointerpretation and can vary between SAR images in a time series, particularly when taken weeks apart. While we agree that this additional information could be beneficial for flood detection, it extends beyond the scope of this work and would require significant additional annotation costs, and a new research and experimental design pathway.
> The primary goal of Kuro Siwo is to enable the development of deep learning algorithms that can accurately isolate and delineate flooded regions, regardless of land cover or waterbody characteristics. Since Kuro Siwo mainly focuses on flash floods, which develop rapidly within a few hours, further refinement of permanent water labels would likely offer marginal benefits. We will clarify this in the manuscript and include waterbody label refinement as a potential area for future work.
>
> Sequential prediction in benchmark:
>
> We agree that the temporal dimension is crucial for flood modelling. BlackBench includes multiple models that leverage this information, such as FloodViT (L279-288) through temporal embeddings and change detection models like FC-EF-Diff, FC-EF-Conc, SNUNet-CD, and Changeformer, which compare pre-event and post-event images. Additionally, inspired also by another reviewer’s comments, we will incorporate ConvLSTM in the camera-ready version, which is a recurrent neural network capable of handling time-series images (refer to the uploaded pdf file for preliminary results).
> While we acknowledge the importance of flood forecasting, this task falls outside the scope of Kuro Siwo. Such efforts would require additional data, such as local weather conditions, water storage capacity, soil information, and denser time series. Kuro Siwo is designed for rapid mapping of flooded areas in flash flood events, driven by excessive rainfall over short periods of time. Mapping flood events that last days or weeks, and are due to compound drivers is not the main focus of Kuro Siwo. This would entail the monitoring of permanent waterbodies for their gradual water extent changes towards the slow and sequential onset of a flood event, which as mentioned in the previous reply, is a different application both in machine learning setup and from a user perspective. Our focus is on supporting emergency response activities within short time frames rather than long-term monitoring of waterbodies for disaster resilience. We will clarify in the manuscript that Kuro Siwo is intended for flash flood mapping and will discuss flood forecasting as a future research direction.

---

### Official Review · Reviewer_d7o8 · 2024-07-25
**A dataset for global flood mapping**

**Rating:** 9
**Confidence:** 5
**Clarity:** The paper is very well written.

**Review:**

This work seems very robust and with high quality. The authors identified a gap in the literature---in this case, in terms of available data---and seek to address it, successfully so. The pros are listed in the strengths section below, and the cons are listed in the opportunities for improvements and limitations.

**Strengths:**

The strengths are: 1) the manually created labels, curated by a team of 5 experts in SAR data; 2) the global coverage of the dataset; 3) the inclusion of SLC data matched to the GRD; and 4) the various models experimented with in the benchmark.

**Additional Feedback:**

Just one minor question:
L178: and one post-event image acquired as close as possible to the actual event date.
	- how is the after determined? after peak flood?

**Correctness:**

I didn't detect any flaw in the design of the dataset nor benchmark, they seem technically sound.

**Documentation:**

The main document and the supplemental material include a high level of detail to insure reproducibility. The supplemental material include links to retrieve the produced dataset. However, I could not locate the links to the benchmark---in the supplemental (L89), it is mentioned the code for the dataset and benchmark are provided via github, but no link is included there.

**Limitations:**

I see no potential negative societal impact. Other than that, the authors have adequately acknowledged the limitations or this work.

**Opportunities For Improvement:**

The authors identified well the limitations and opportunities for improvement,  which can be seem as future work. I think the one which would be more tangible for this specific publication is the assessment of the SLC dataset on par with the other experiments in the benchmark. More specifically, the authors report on the accuracy of some early results with the SLC dataset, but it is unclear how those results would compare with the rest of the experiments in the benchmark (Table 2). To be clear, that would be a nice addition to this work, but I don't see it as requirement for acceptance of this work.

**Relation To Prior Work:**

Yes, this work is well contextualized.

**Summary And Contributions:**

This paper presents a novel SAR dataset, with data captured from Sentinel-1. This dataset is primed for machine learning tasks of global flood mapping, as it is incorporates manually created labels over diverse regions and climate zones.

---

> ### Author Rebuttal · Authors · 2024-08-17
>
> We thank the reviewer for recognizing the quality of our work and for the valuable suggestions.
>
> Assessment of the SLC dataset:
>
> We initially limited the assessment of the SLC dataset to avoid extensive large-scale experiments, as this work does not focus on methods specifically designed to exploit complex SLC data. However, we acknowledge the importance of providing baselines for future research on SLC. To address this, we have trained most of the BlackBench models on the complete SLC dataset and will include the full SLC benchmark results in the supplementary materials for the camera-ready version. Preliminary results (see Table 2 in the attached pdf file) indicate that our top-performing model for the SLC benchmark, UPerNet with a Swin-S backbone, achieves 70.39% F1-Score for the flood class, 79.48% F1-Score for the permanent water class, and 78.17% F1-Score for the binary water/no-water task. The second-best model, UPerNet with a Swin-B backbone, shows comparable performance on the binary task with a 77.23% F1-Score and 70.25% F1-Score for the permanent water class, and 79.16% F1-Score for the flood class. Although these models perform well in detecting water bodies, they tend to prioritise one of the two water classes more than what was observed with the GRD dataset. This is expected as the GRD preprocessing pipeline is optimised for flood mapping, while our SLC models used standard architectures not specifically tailored to leverage the detailed complex valued information provided by SLC data.
>
> Code for the benchmark:
>
> Concerning the benchmarking, we have uploaded all necessary code in a zip file (please refer to the link given to the AC in the official comment). In the camera-ready version of the paper, all anonymized Dropbox links will be replaced with the official GitHub repository.
>
> Determination of event date:
>
> Kuro Siwo focuses on developing methods for the rapid delineation of flash flood events. For CEMS-activated events, the reference date corresponds to the date provided by CEMS, which might be the peak water levels or another relevant date depending on the user activation requirements. For non-CEMS events, the reference date is typically aligned with the peak water levels. Generally, in both scenarios, this reference date closely coincides with both the start of the flood event and its peak extent, as flash floods usually develop within hours following heavy rainfall or other sudden excess water releases, such as dam failures or hurricanes.

---

> > ### Comment · Reviewer_d7o8 · 2024-08-30
> > **good work**
> >
> > I am very pleased with the answers provided to my comments, as well as to all the other reviewers' comments. I hope all the details added in the responses can be included in the final manuscript (main doc or supp.)
> > I am confident this is a top submission and hold my scores as such.

---

### Official Review · Reviewer_b48H · 2024-08-08
**Kuro Siwo**

**Rating:** 7
**Confidence:** 3
**Clarity:** The paper is well written.

**Review:**

Kuro Siwo is a new dataset that introduces manually annotated Sentinel-1 synthetic aperture radar (SAR) images for 43 global flood events from 2015 to 2022. It includes a large unlabeled set of SAR samples and BlackBench and extensive benchmarking.

As seen in Figure 1, the spatial distribution of global flood events is highly concentrated in certain areas, such as Europe. How were those events picked? Why was a more even distribution not considered?

What is the intent of the use of SLC data?

Why not incorporate S2 as well? In times/conditions with a break in clouds, S2 produces more accurate flood maps than S1. The authors only mention the good things about using S1 for flood mapping, but that is impartial, so I suggest that they list the challenges associated with mapping floods with S1 as well.

I see the reason behind the dataset's name. It could be better to use something more intuitive, a name with Floods in it, e.g., S1Floods.

The elevation is typically important when mapping water/floods. Could it be that it wasn't important because the authors didn't use a higher-grade DEM?

Given the triplet images picked per flood and the only ascending or descending rule imposed and sensor overpass, what was, on average, the number of days post-flood event the acquisition of the 3rd S1 image? This could be very impactful if not the same day as the flood event. The authors do not give any number for this critical aspect.

The authors had difficulty distinguishing flooded areas from permanent water in some cases. Did they consider using an existing permanent water layer as the input? If yes, which one?

How were the 5 SAR experts picked? What qualified as a SAR expert for the purpose of this work?

**Strengths:**

Extensive benchmarking.

**Additional Feedback:**

NA

**Correctness:**

It seems sound, although I'd like to see a section addressing the cons of using S1 for flood mapping, especially in, say, urban areas. I believe mostly the pros were mentioned in the paper.

**Documentation:**

yes

**Ethics:**

No concerns

**Limitations:**

Mostly, yes.

**Opportunities For Improvement:**

Answering the Qs I posed above.

**Relation To Prior Work:**

I'm not sure how important it is, but it looks like a previous version of the dataset was available before. Is there a difference between the two other than the number of floods? Kuro siwo: 12.1 billion m2 under the water. a global multi-temporal satellite dataset for rapid flood mapping. arXiv preprint arXiv:2311.12056, 2023.

**Summary And Contributions:**

The paper presents a new annotated flood dataset based on Sentinel 1 data. It covers 43 global flood events, and 5 SAR experts annotated it. The paper presents an extensive benchmark, which is great.

---

> ### Author Rebuttal · Authors · 2024-08-17
>
> We thank the reviewer for their comments and suggestions. We address each point in detail below:
>
> Events selection:
>
> As noted in L197, the events in Kuro Siwo are divided into two categories: a) those with available CEMS annotations and b) those without. Refining existing CEMS annotations significantly accelerated the process compared to manually digitising flooded and permanent water areas from scratch (L209-229). However, since CEMS focuses on Europe (L230-233), we extended our dataset to achieve broader global coverage by incorporating major flood events from other continents. These events extend to large areas offering a significant number of training patches, compensating for the imbalanced amount of events. This means that while e.g. Europe has more events (23) compared to Asia (5), the latter contributes with more samples (26,605 vs 22,803).
>
> Importance of SLC data:
>
> Our decision to include the SLC component in the dataset is driven by three main observations. First, Kuro Siwo leverages the phase information provided by the SLC data, which can be critical for many applications through phase statistics, signal decorrelation and coherence maps (L127-134). Also, providing both amplitude and phase information in the SLC component enables the development of novel methods for complex-valued inputs. Finally, representation learning through the unlabelled component of Kuro Siwo significantly increases the range of potential applications. The SLC data offer flexibility for users to apply their own preprocessing pipelines tailored to any downstream task. Our motivations for including SLC data are further discussed in L190-196.
>
> Incorporating S2 data:
>
> We agree that S2 data can significantly aid flood mapping under cloud-free conditions. Our focus on SAR data was driven by the need for reliable, all-weather systems (L49-56). Incorporating S2 would require extensive photointerpretation, as it captures imagery at different times than S1, often leading to discrepancies in detected flood extents, since flash floods have a short life cycle. Nonetheless, the synergistic use of both datasets is a promising research direction, combining the precision of multispectral data with the all-weather resilience of SAR. Having Kuro Siwo as a publicly available resource with reliable SAR based annotations can provide the foundation for assembling such a multi-modal dataset. We plan to highlight this as potential future work in the camera-ready version.
>
> Challenges associated with mapping floods with S1:
>
> We thank the reviewer for the important suggestion and will expand the limitations section for the camera-ready version. For example, speckle effect introduces granular noise that complicates the differentiation between flooded and non-flooded areas. Complex terrain and varying land cover types, such as forests, further complicate flood detection by affecting radar signal interaction. Variations in water surface roughness due to wind, or vegetation can alter backscatter, making it difficult to consistently identify floodwaters. Detecting floods beneath dense vegetation is also challenging, as the S1 C-band radar signal may not penetrate sufficiently (as opposed to L-band SAR sensors) to reveal underlying water. Urban areas pose another difficulty, as radar signals often produce complex scattering effects (e.g. double-bounce reflections), making it difficult to distinguish between water and other surfaces, like wet roads or buildings. Finally, the limited temporal resolution of S1, with revisit times of 12 days (after the malfunctioning of S1-B), can result in missing critical changes during dynamic flood events.
>
> DEM selection:
>
> We used the freely available SRTM 1Sec DEM, which offers a 30m GSD and global coverage and does not impose additional financial costs on users. Although we did not test commercial higher-grade DEMs, we hypothesise that while improved DEM resolution could enhance model performance, the 10m GSD of Sentinel-1 data may limit such gains.
>
> Number of days between event and post-flood image:
>
> On average, the third S1 image was captured 3.6 days post-event (with std of 6.07 days), and the median delay was 1 day. We'll include this in the camera ready. Acquisition is constrained by orbit direction and satellite overpass. Revisit rate also decreases closer to the equator, leading to longer gaps in some regions, e.g. Myanmar, Togo, etc.
>
> Permanent water layer as input:
>
> Permanent water bodies are dynamic and change over time due to droughts, sediment transportation, etc. In our initial experiments, we observed discrepancies when using the Open Street Map water layer, which led us to prefer manual photointerpretation of permanent water bodies. Therefore, using an existing water layer as input to the models could potentially impute noise and ultimately disrupt the learning process. However, its exact impact remains an open question. We will add this point to the discussion section for the camera ready.
>
> Selection of the 5 SAR experts:
>
> Our team comprised professionals and researchers with expertise in SAR image characteristics, analysis and photointerpretation, with emphasis on speckle, layover/foreshortening/shadow effects and surface scattering attributes. The team leader is a Professor in Remote Sensing with a PhD in SAR remote sensing, who has also managed several flood delineations for CEMS activations. The team also included experienced researchers (a geographer and a civil engineer) with over 10 years of experience in SAR and CEMS, as well as two researchers specialising in Remote Sensing and GIS applications. The annotation process was further standardised as discussed in Section 4 of the suppl. material.
>
> Prior Work:
>
> Kuro Siwo was initially released as a preprint on arXiv and has since been significantly improved for this NeurIPS submission. We expanded the number of events to include more floods outside Europe, added aligned SLC data and enhanced FloodViT with temporal information.

---

### Author Rebuttal · Authors · 2024-08-17

We sincerely appreciate the reviewers' constructive feedback and comments, which have significantly contributed to improving the manuscript's quality. We are pleased that our work has been well received and described as “very robust with high quality.” We believe this recognition underscores the potential of our research to advance the field of flash flood mapping, a critical area given the increasing frequency and impact of such events. The key contributions of our work are as follows:

1) A unique global, large-scale SAR dataset with meticulously curated manual annotations.
2) The inclusion of both GRD and SLC Synthetic Aperture Radar products, which enables the end user to adopt any preprocessing pipeline suitable to their needs.
3) An extensive benchmark, BlackBench, offering strong baselines that can guide and inspire future methodologies.

We tried to address all the questions and concerns raised by the reviewers within the individual threads. Below is a summary of the key actions we have taken to incorporate the reviewers’ suggestions:

1) We have provided detailed information on the number of labelled and unlabelled samples in the Kuro Siwo dataset. Please refer to Table 1 in the attached pdf file for this clarification.
2) We expanded the benchmark for the full SLC dataset including all models that were used in the GRD component of Kuro Siwo and propose to include it in the supplementary material. Preliminary results of the extended benchmark can be found in Table 2 of the attached pdf file.
3) To ensure comprehensive coverage of model architectures, we have added ConvLSTM [1] to the BlackBench models as a robust representative of recurrent neural network architectures. The results are detailed in Tables 2 and 3 of the attached pdf file.
4) For completeness, we have uploaded the code for BlackBench in a zip file under an official comment here. Additionally, all supplementary material links will be updated for the camera-ready version to point to the official public GitHub repository, which will contain all the code and data used in this work.

Furthermore, we would like to summarise our responses to a few crucial questions raised by some reviewers:

Importance of the SLC component

Even though the GRD component is optimised for flash flood mapping, it sacrifices phase information— phase statistics, signal decorrelation and coherence maps can be used in tasks like change detection, particularly being a good driver for flood segmentation. As discussed in L190-196, the introduction of the SLC data provides both amplitude and phase information enabling the investigation of future methods that exploit this complex valued input. Even more, taking into consideration the unlabelled component of Kuro Siwo, the potential for representation learning is greatly expanded, opening up a wide array of applications. By providing SLC data with minimal preprocessing, users can tailor Kuro Siwo to their specific downstream tasks by applying their own preprocessing pipeline, without affecting the integrity of the dataset, which significantly adds to the flexibility of Kuro Siwo. This approach frees up pre-processing restrictions applicable for the GRD-based pipeline, enhances the dataset's versatility and broadens its applicability across different research domains.

Over-representation of Europe in Kuro Siwo

As we discuss in Section 3 (L197), the events in Kuro Siwo are categorised in two groups: a) those with available Copernicus Emergency Management Service (CEMS) annotations and b) those without. Refining existing CEMS annotations significantly accelerated the process compared to manually digitising flooded and permanent water areas from scratch, as explained in L209-229. However, given CEMS focus on Europe (L230-233), we expanded our dataset to achieve broader global coverage by incorporating major flood events, annotated from scratch, from Asia, North and South America, Australia, and Africa. These events cover extensive areas providing a substantial number of additional training patches. Floods in Europe are fragmented small scale events, as opposed to large floods events encountered in other continents, e.g. Asia. This means that while most events in Kuro Siwo are from Europe wrt to Asia (23 out of 47 vs 5 out of 47), most flood samples in Kuro Siwo are from Asia (26,605 in Asia vs 22,803 in Europe). Our main objective was to include as many training samples as possible from all 6 continents, given the restrictions imposed by the varying S1 revisit times and the availability of useful post-flood captions.

Utilisation of temporal information in the benchmark

BlackBench explicitly models temporal information through various methods. Change detection models handle pre-event and post-event captions separately, extracting useful semantic information as changes evolve through time, while FloodViT, as described in L283-288, leverages temporal embeddings for each timestep. These embeddings are generated by independently encoding the year, month, and day of the caption using a sinusoidal encoding and are subsequently combined with learnable positional embeddings. The encoded tokens are then fed to the transformer, thus  important temporal information is imputed to the learning process.
In addition, we have now included ConvLSTM, a recurrent neural network that achieved promising results with 77.53% F1-Score for flood and 76.57% for permanent waters, in the camera-ready version (see Table 3 in the attached pdf). While we acknowledge there are multiple ways to model time series, our goal is not to provide a new SOTA for flood mapping but to present a quality dataset that fosters the development of such SOTA methods. BlackBench provides a strong benchmark and robust baselines to support this.

[1] Shi, Xingjian, et al. "Convolutional LSTM network: A machine learning approach for precipitation nowcasting." Advances in neural information processing systems 28 (2015).

---

### Decision · Program_Chairs · 2024-09-26

**Decision:**

Accept (Poster)

**Comment:**

The paper is appreciated by two reviewers and two are less enthusiastic. Nevertheless, the two lukewarm reviews are very short and vague in their requests and moreover the reviewers chose not to participate to the discussions. Therefore I decided to weigh in the extensive rebuttals and the positive reaction be the two other reviewers to comments and to accept the paper. congratulations!